# Development and Testing of the A1 Volumetric Air Sampler, an Automatic Pollen Trap Suitable for Long-Term Monitoring of eDNA Pollen Diversity

**DOI:** 10.3390/s22176512

**Published:** 2022-08-29

**Authors:** Gulzar Khan, Albrecht Hegge, Birgit Gemeinholzer

**Affiliations:** 1AG Botanik, University of Kassel, Heinrich-Plett-Str. 40, 34132 Kassel, Germany; 2A1 Productdesign, Reindl + Partner GmbH, Bahnhofstraße 13, 50999 Cologne, Germany

**Keywords:** eDNA monitoring, trap robot, biodiversity monitoring, pollen, pollen traps

## Abstract

Airborne pollen surveys provide information on various aspects of biodiversity and human health monitoring. Such surveys are typically conducted using the Burkard Multi-Vial Cyclone Sampler, but have to be technically optimized for eDNA barcoding. We here developed and tested a new airborne pollen trap, especially suitable for autonomous eDNA-metabarcoding analyses, called the A1 volumetric air sampler. The trap can sample pollen in 24 different tubes with flexible intervals, allowing it to operate independently in the field for a certain amount of time. We compared the efficiency of the new A1 volumetric air sampler with another automated volumetric spore trap, the Burkard Multi-Vial Cyclone Sampler, which features shorter and fewer sampling intervals to evaluate the comparability of ambient pollen concentrations. In a sterile laboratory environment, we compared trap performances between the automated volumetric air samplers by using pure dry pollen of three species—*Fagus sylvatica*, *Helianthus annuus* and *Zea mays*—which differ both by exine ornamentation and pollen size. The traps had a standard suction flow rate of 16.5 L/min, and we counted the inhaled pollen microscopically after a predefined time interval. Our results showed that though we put three different pollen types in the same container, both the traps inhaled all the pollens in a statistically significant manner irrespective of their size. We found that, on average, both traps inhaled equal an number of pollens for each species. We did not detect any cross-contamination between tubes. We concluded that the A1 volumetric air sampler has the potential to be used for longer and more flexible sampling intervals in the wild, suitable for autonomous monitoring of eDNA pollen diversity.

## 1. Introduction

Airborne pollen monitoring is useful for different health-related issues (e.g., allergy) [1], GMO [2] and vegetation monitoring [3,4]. Traditionally, airborne pollen monitoring has been accomplished using different passive/gravimetric volumetric samplers (samplers without any air suction pumps) [5,6,7,8,9]. Today, volumetric pollen samplers are mainly based on the Hirst spore trap (provided with a suction pump) [2,5,10,11]. The efficiencies of volumetric samplers are generally high; however, they depend on particle size, and wind velocity [11]. Pollen analysis is then achieved by microscopic identification, which demands expert knowledge and is time consuming [12,13,14].

Emerging techniques are DNA-based pollen monitoring and species identification (called eDNA barcoding or metabarcoding) that do not depend on microscopic identification and, when standardized, they can be highly automated and standardized. It has been proven that the method is highly efficient and sensitive [15,16,17,18,19]. DNA extracted from the environment (eDNA), amplification of specific eDNA barcoding regions via PCR and high-throughput sequencing (NGS) provide sequences, that—when compared to reference databases of known species and sequences—can reveal information about the diversity of organisms in an environmental sample. eDNA-based research is becoming increasingly important as it provides information not only about plant traces but also on the biotic composition of entire ecosystems [16,20].

Nevertheless, the most important step in eDNA metabarcoding is to sample the air-borne pollen in an efficient way. Traditional volumetric pollen collectors such as the Hirst spore trap need to be equipped with vessels instead of tapes to capture airborne plant traces for metabarcoding analyses. In addition, different control options for multiple vessels would be desirable to provide greater temporal flexibility for different collection intervals. Currently, the only volumetric air sampler with the ability to use vessels instead of tapes is the Burkard Multi-Vial Cyclone Sampler (Burkard Manufacturing Co. Ltd., Rickmansworth, UK). This trap is equipped with a rotating carousal, with seven vials that allow sampling for seven days (each vial for a maximum of 24 h) through a motor and control system. We developed a new Hirst-type automatic pollen trap, called the A1 volumetric air sampler, which collects also in vials, similar to the Burkard Multi-Vial Cyclone Sampler, however, with additional features.

The A1 volumetric air sampler can collect pollen in 24 × 2 mL tubes at controlled sampling intervals with a forward as well as backward control system. This allows interval flexibility, e.g., for multiple day–night sampling or daily/weekly/bi-weekly intervals for autonomous sampling. The A1 volumetric air sampler can work with battery, solar-power as well as a permanent power connection (Appendix A). This air sampler has extendable legs and a watermark level, allowing stable setup even on uneven terrain. Different tests have been conducted in controlled environments to check for cross-contamination between vials of the A1 volumetric air sampler and compared to the one of the Burkard Multi-Vial Cyclone Sampler with pollen of different sizes and exine ornamentation. The experiment was based on the manual default cyclone setting of both traps in order to exclude the slightly different parameters of the adjustable programs as an influencing factor. We asked two questions to check if: (I) we detect cross-contamination from the cyclone and lid to the tubes as well as between the tubes and (II) whether there are significant differences in collecting pollens/spores between the Burkard Multi-Vial Cyclone Sampler and the A1 volumetric air sampler. We also investigated some follow-up hypotheses, such as whether the two machines behaved differently when provided with pure pollen in a mixture, and whether sampling was influenced by small, medium, or larger pollen.

## 2. Materials and Methods

### 2.1. Experimental Design

We conducted tests in a controlled environment using pure dry pollens (Bonapol, Czech Republic). Prior to usage, pollens were stored in closed containers to exclude contamination and were handled under a sterile bench during the experiments to avoid cross-contamination. Three different pollen types were tested: larger ones (~80 µm diameter, *Zea mays*; Poaceae; pollen monocolpate and rugulate), medium-sized ones (~50 µm diameter, *Fagus sylvatica*; Fagaceae; pollen tricolpate, from scabrate to rugulate), and smaller-sized ones (~20 µm diameter, *Helianthus annuus*; Asteraceae; tricolpate and echinate) with different exine ornamentation (Appendix A).

The A1 volumetric air sampler and the Burkard Multi-Vial Cyclone Sampler were attached to power cables since the air circulation of both traps was subject to fluctuations. We followed the experiment using the manual default cyclone setting of both traps (air flow rate 16.5 L/min) to be able to exclude the slightly different parameters of the adjustable programs of both traps as an influencing factor. Containers were attached to the opening of both traps. T-shaped openings were cut into the container and connected to an air source to provide an approximately equal air flow for each trap and to ensure air circulation (Appendix A). Two closed 16.5 L bags were connected to the T-slot and provided with the air source for one minute. This step is important to swirl the ambient air in the closed container to ensure that the sampling behavior of the traps are equal. After tuning the air flow in alignment with the trap’s suction air pressure, equal amounts of pure dried medicinal pollen stored in the containers were added (Appendix A), while the traps and air flow had simultaneously started.

Preliminary tests with different time intervals (5 min, 4 min, 3 min, 2 min, 1 min, and 30 s) showed that pollens were completely succeeded in (or at least disappeared from the container to the visible eyes) from the closed containers into the trap tubes within 30 s irrespective of their volume. After ensuring uniform air sucking success, we performed different tests. We took equal pollen volumes of the three different pollen types individually and in a mixture and placed them in the containers through the opening and ran the multi-vial pollen samplers for comparison. The pollen mixes were collected in 2 mL (A1 volumetric air sampler) and 1.5 mL (Burkard Multi-Vial Cyclone Sampler) tubes, respectively, with the manual fan setting of 100% fan cycling and 30 s. To check for contamination, two sterile tubes before and after the collection tubes of each trap were also analyzed for pollen content. The experiment was repeated five times.

### 2.2. Microscopy and Statistical Analyses

For microscopy, we added 1000 mL of ddH_2_O to each tube and shook vigorously to prepare a homogeneous pollen suspension. Respectively, 10 µL of the suspensions was taken and placed on the slide of a hemocytometer, and the pollen in each hemocytometer field were counted. For each tube, 10 replicates of the count were performed. The averages of all counts were calculated and multiplied by 100 (Table 1). No pollen was ever found in the test tubes adjacent to the collection containers, so cross-contamination can be excluded.

To test the statistical difference in pollen sampling between the two traps, the Wilcoxon pairwise comparison test was applied for each pollen species and the mixtures [21]. It is a non-parametric rank sum test which compares two paired groups and calculates the difference between their medians. The test considers that both medians are significantly different when the *p* is less than the specified threshold (details see Appendix A). To statistically calculate whether each trap captured the same number of pollens from the mixture, we use Analysis of Variance (ANOVA) and the Kruskal–Wallis test. Both tests are useful when there are three or more groups. Significance was further refined using the Bonferroni correction with 95% confidence intervals (Appendix A). The results are plotted in violin plots using the plot package from tidyverse. ALL analyses were executed in R studio [22,23].

## 3. Results and Discussion

We found that the highest number of pollens inhaled by both the Burkard Multi-Vial Cyclone Sampler and the A1 volumetric air sampler were for the species *Helianthus annuus*, with smallest size and echinate exine ornamentation. Similar, results have been revealed in previous studies, e.g., [24]. In our treatment with 60 s and 100% fan cycling, we retrieved an average of 18,974 (range 8670–30,070) pollens with the Burkard Multi-Vial Cyclone Sampler and 16,362 (range 1850–28,930) with the A1 volumetric air sampler. Similarly, the smallest number of pollens retrieved using both traps were for the larger pollens *Zea mays* (~80 µ in diameter), 1610 (range 650–3260) by Burkard Multi-Vial Cyclone Sampler and 1838 (range 260–3260) by the A1 volumetric air sampler. *Fagus sylvatica* placed in the middle (Table 2, Figure 1). The results suggest that both traps inhaled smaller pollen in higher number. The Wilcoxon test revealed that both traps of each species do not have significant difference (*p* = 1) in their average medians, which reflect that both are equal in performance, especially in the case of inhalation of the same size and exine structures. This is in concordance with earlier studies which revealed that the Burkard Multi-Vial Cyclone Sampler inhale the pollens of different shapes in different number [7,13,25,26,27].

After comparing the pollen counts in the tubes in relation to each trap, ANOVA and the Kruskal–Wallis test revealed non-significant differences in the average median of the A1 volumetric air sampler (Table 2), while this was not the case for the Burkard Multi-Vial Cyclone Sampler, except for *F. sylvatica* and *Zea mays* (Table 2). Similarly, the global *p*-values for the A1 volumetric air sampler trap were not statistically significant, while they were highly significant for the Burkard Multi-Vial Cyclone Sampler (Table 2). Lu et al. [24] found same trend using the Burkard Multi-Vial Cyclone Sampler. They retrieved more small-sized pollen of *Artemisia* (<10–25 µm) [28] than larger-sized Poaceae species in a study about air-borne pollen concentrations. These results are in concordance with previous studies [13,24,26].

## 4. Conclusions

We compared two automatic pollen traps suitable for long-term monitoring of eDNA pollen diversity, the Burkard Multi-Vial Cyclone Sampler, and the A1 volumetric air sampler. Both traps are suitable for long-term monitoring of pollen, plant traces and other traces of eDNA (environmental DNA) but feature different sampling intervals and tube capacities. Our results confirm equal efficiency and suitability for long-term monitoring activities and constructional similarities to the Hirst volumetric air sampler, most likely resulting in similar pollen collection efficiencies. We concluded that the A1 volumetric air sampler can be used as an alternative to the Burkard Multi-Vial Cyclone Sampler, providing collection capabilities beyond that if needed. Additionally, the A1 volumetric air sampler has the potential to be used autonomously for a period of up to one year in more flexible sampling intervals in the wild.

## Figures and Tables

**Figure 1 sensors-22-06512-f001:**
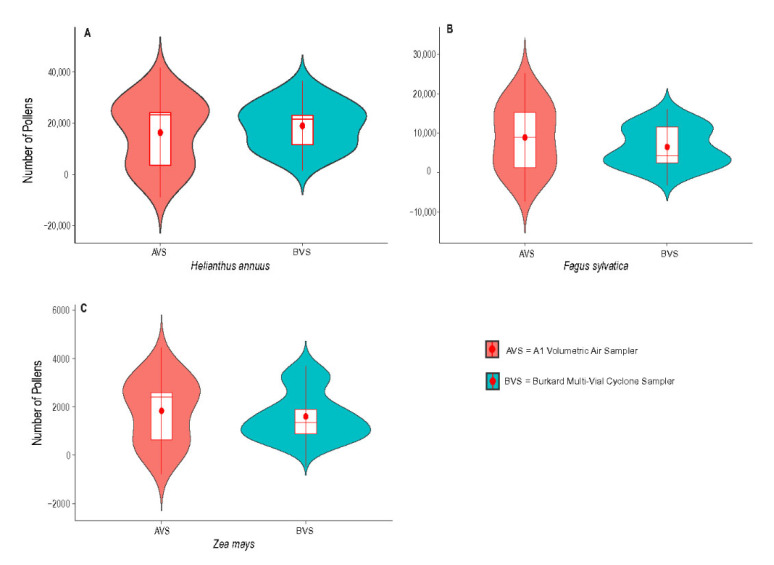
Pairwise comparison of pollens inhaled by the A1 volumetric air sampler and the Burkard Multi-Vial Cyclone Sampler for (**A**) *Helianthus annuus*, (**B**) *Fagus sylvatica* and (**C**) *Zea mays*.

**Table 1 sensors-22-06512-t001:** Details of pollen counting.

Repeats	COUNTS PER TUBE										Average	Total
*Fagus sylvatica*/Burkard Multi-Vial Cyclone Sampler												
	1	2	3	4	5	6	7	8	9	10		
1	120	125	103	109	94	131	100	128	150	127	118.7	11,870
2	107	109	151	158	90	101	126	90	105	120	115.7	11,570
3	9	10	15	22	65	12	23	56	31	12	25.5	2550
4	40	46	13	86	27	135	15	11	26	37	43.6	4360
5	11	13	23	9	14	17	14	14	46	70	23.1	2310
*Helianthus annus*/Burkard Multi-Vial Cyclone Sampler												
	1	2	3	4	5	6	7	8	9	10		
1	224	270	230	310	391	305	363	350	301	263	300.7	30,070
2	243	206	276	292	203	198	225	227	170	203	224.3	22,430
3	51	50	56	84	374	43	72	224	144	67	116.5	11,650
4	273	194	50	371	105	872	37	25	171	53	215.1	21,510
5	57	43	87	23	122	37	48	91	153	206	86.7	8670
*Zea mays*/Burkard Multi-Vial Cyclone Sampler												
1	12	17	18	19	11	26	14	22	30	20	18.9	1890
2	48	46	60	32	31	34	26	22	15	12	32.6	3260
3	3	5	7	6	9	5	4	11	12	3	6.5	650
4	15	8	4	13	7	47	7	9	15	11	13.6	1360
5	3	5	11	3	6	9	5	7	17	23	8.9	890
*Fagus sylvatica*/A1 volumetric air sampler												
1	130	106	121	150	163	153	158	207	172	173	153.3	15,330
2	12	38	35	1200	393	17	37	40	39	41	185.2	18,520
3	1	1	2	1	2	5	15	4	12	15	5.8	580
4	2	4	3	15	13	11	9	11	25	29	12.2	1220
5	133	73	113	177	130	87	40	53	47	40	89.3	8930
*Helianthus annuus*/A1 volumetric air sampler												
1	225	211	283	223	230	257	242	281	191	263	240.6	24,060
2	27	81	90	1700	683	33	67	62	85	65	289.3	28,930
3	2	3	4	1	4	20	43	11	43	53	18.4	1840
4	10	12	17	21	15	23	33	122	50	65	36.8	3680
5	100	240	200	430	398	175	90	87	117	93	193	19,300
*Zea mays*/A1 volumetric air sampler												
1	21	13	40	25	26	37	23	36	23	16	26	2600
2	7	12	11	120	29	13	13	9	11	17	24.2	2420
3	1	1	1	1	1	3	7	2	5	4	2.6	260
4	1	3	1	5	4	7	5	17	13	9	6.5	650
5	50	57	30	66	33	23	10	15	19	23	32.6	3260

**Table 2 sensors-22-06512-t002:** ANOVA and the Kruskal–Wallis test for pollen capture differences in the A1 volumetric air sampler and the Burkard Multi-Vial Cyclone Sampler. HA: *Helianthus annuus*; FS: *Fagus sylvatica*; ZM: *Zea mays*; *p*: significant difference in pollen capture species wise; significance *p* < 0.05; *: significant and **: highly significant.

Species	*p* Based on Global ANOVA	*p* Based on Global Kruskal–Wallis Test	*p* Based on Kruskal–Wallis Test		
			HA vs. FS	HA vs. ZM	FS vs. ZM
A1 wind trap	0.06	0.9	0.6	0.06	0.7
Burkard spore trap	0.00 **	0.00 **	0.02 *	0.00 **	0.6

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
