# Peer review of "Development and Testing of the A1 Volumetric Air Sampler, an Automatic Pollen Trap Suitable for Long-Term Monitoring of eDNA Pollen Diversity"

_sensors, 2022, doi:10.3390/s22176512_

Round 1

Reviewer 1 Report

The present paper reviews the development of a new automatic pollen trap for monitoring eDNA pollen diversity, which is called A1 volumetric air sampler. More specifically, this sampler has 24 different tubes with flexible intervals and can be used autonomously for monitoring for a period of up to one year in the wild. It inhales pollen of different size microscopically and provides biological information for the aerial environment. This study has examined and compared the efficiency of this new pollen trap with another volumetric spore trap by using three different pollen species. Both samplers present the same trap performance and inhale equal number of pollens of each species, but the new one has been proven more suitable for longer monitoring of eDNA pollen diversity.

It’s a well-organized experimental study. The method and the results are well described. The manuscript is clear. Figures are appropriate and easy to understand, but they need corrections regarding their classification. Tables should be presented according to the instructions of the Journal. The supplementary information which is referred in line 133 is not presented in the manuscript or in a different file. Some revisions should be performed concerning mostly the English language. The whole text is written in the first plural person, and it should be converted to third person. All references should be corrected and written according to Journal’s instructions.

An itemized comment list follows.

Author Response

Thanks a lot for your time and efforts to make this work more attractive. Please see for the complete details the attached file in .docx. we already incorporated the changes as suggested in the manuscript

Reviewer 2 Report

Testing different pollen trap designs is a great start. For Burkard 7-day Volumetric Spore Trap this has been done and several papers have been cited in this manuscript. But cyclone-type traps are clearly insufficiently tested. But there are examples:

Alcázar P, Galán C, Torres C, Domínguez-Vilches E. Detection of airborne allergen (Pla a 1) in relation to Platanus pollen in Córdoba, South Spain. Ann Agric Environ Med. 2015; 22(1): 96–101. doi: 10.5604/12321966.1141376

Parker, M. L., McDonald, M. R., and Boland, G. J. 2014. Evaluation of air sampling and detection methods to quantify airborne ascospores of Sclerotinia sclerotiorum. Plant Dis. 98:32-42.

There are also fresh reviews on all types of traps:

Gediminas Mainelis (2020) Bioaerosol sampling: Classical approaches, advances, and perspectives, Aerosol Science and Technology, 54:5, 496-519, DOI: 10.1080/02786826.2019.1671950

I hope this work is the start of a major study to standardize and test automated aeroallergen eDNA detection systems. And this promising sampler will be tested in a real environment and will be used to do work on allergen eDNA detection.

However, in its present form, the work requires a lot of improvement.

1. It is necessary to describe the pattern of the experiment more clearly.

2. In addition to the photo of the device, add a diagram or block diagram of the experiment and the test device.

3. The materials and methods say: For this we took equal pollen volumes of the three different pollen types individually and in a mixture and placed them in the containers through the opening and ran the multi vial pollen samplers for comparison. What does this mean? Testing order for different pollen types or testing with pollen mix?

4. In numbers and measurements, you need to check for typos.

5. In table S1, the amount of pollen in Burkard Multi-Vial Cyclone Sampler tubes 1 to 5 decreases. In test tubes A1, the minimum of pollen is 3-4 tubes, and in the last one there is always a lot. Why then!

6. Literature cited off topic. All the time cited are works made by Burkard 7-day Volumetric Spore Trap. It has nothing to do with cyclone-type traps! The literature needs to be found and analyzed again.

Detailed comments are in the file.

Author Response

we mostly complied with the comments in this revised version of the manuscript, hope these will be enough to cover the scope of this work.
